# Altered Phosphorylation of Cytoskeleton Proteins in Peripheral Blood Mononuclear Cells Characterizes Chronic Antibody-Mediated Rejection in Kidney Transplantation

**DOI:** 10.3390/ijms21186509

**Published:** 2020-09-05

**Authors:** Maria Teresa Rocchetti, Federica Rascio, Giuseppe Castellano, Marco Fiorentino, Giuseppe Stefano Netti, Federica Spadaccino, Elena Ranieri, Anna Gallone, Loreto Gesualdo, Giovanni Stallone, Paola Pontrelli, Giuseppe Grandaliano

**Affiliations:** 1Clinical Pathology Unit and Center for Molecular Medicine, Department of Medical and Surgical Sciences, Faculty of Medicine, University of Foggia, 71122 Foggia, Italy; mariateresa.rocchetti@unifg.it (M.T.R.); federica.rascio@gmail.com (F.R.); giuseppestefano.netti@unifg.it (G.S.N.); federica.spadaccino@unifg.it (F.S.); elena.ranieri@unifg.it (E.R.); 2Nephrology Dialysis and Transplantation Unit, Department of Medical and Surgical Sciences, Faculty of Medicine University of Foggia, 71122 Foggia, Italy; giuseppe.castellano@unifg.it (G.C.); giovanni.stallone@unifg.it (G.S.); 3Nephrology Dialysis and Transplantation Unit, Department of Emergency and Organ Transplantation, Faculty of Medicine, University of Bari “Aldo Moro”, 70124 Bari, Italy; mfiorentino84@gmail.com (M.F.); loreto.gesualdo@uniba.it (L.G.); 4Experimental Biology, Department of Basic Medical Sciences, Neuroscience and Sense Organs, Faculty of Medicine, University of Bari “Aldo Moro”, 70124 Bari, Italy; anna.gallone@uniba.it; 5Experimental Biology, Department of Emergency and Organ Transplantation, Faculty of Medicine, University of Bari “Aldo Moro”, 70124 Bari, Italy; 6Nephrology Unit, Department of Medical and Surgical Sciences, Fondazione Policlinico Universitario “A. Gemelli” IRCCS, 00168 Rome, Italy; giuseppe.grandaliano@unicatt.it; 7Department of Translational Medicine and Surgery, Faculty of Medicine, Catholic University of the Sacred Heart, 00168 Rome, Italy

**Keywords:** chronic antibody-mediated rejection, phosphoproteome, actin-related protein 2, F-actin, kidney transplantation

## Abstract

Chronic antibody-mediated rejection (CAMR) is the major cause of kidney transplant failure. The molecular mechanisms underlying this event are still poorly defined and this lack of knowledge deeply influences the potential therapeutic strategies. The aim of our study was to analyze the phosphoproteome of peripheral blood mononuclear cells (PBMCs), to identify cellular signaling networks differentially activated in CAMR. Phosphoproteins isolated from PBMCs of biopsy proven CAMR, kidney transplant recipients with normal graft function and histology and healthy immunocompetent individuals, have been investigated by proteomic analysis. Phosphoproteomic results were confirmed by Western blot and PBMCs’ confocal microscopy analyses. Overall, 38 PBMCs samples were analyzed. A differential analysis of PBMCs’ phosphoproteomes revealed an increase of lactotransferrin, actin-related protein 2 (ARPC2) and calgranulin-B in antibody-mediated rejection patients, compared to controls. Increased expression of phosphorylated ARPC2 and its correlation to F-actin filaments were confirmed in CAMR patients. Our results are the first evidence of altered cytoskeleton organization in circulating immune cells of CAMR patients. The increased expression of phosphorylated ARPC2 found in the PBMCs of our patients, and its association with derangement of F-actin filaments, might suggest that proteins regulating actin dynamics in immune cells could be involved in the mechanism of CAMR of kidney grafts.

## 1. Introduction

The innovation of immunosuppressive protocols in the last four decades significantly reduced the incidence and impact of acute cellular rejection in kidney transplantation. However, we are still facing the challenge of reduced long-term survival of kidney grafts and, as suggested by Sellares et al., in this relatively new scenario chronic antibody-mediated rejection is the major cause of kidney transplant failure [1,2]. This condition is clinically characterized by proteinuria and a progressive deterioration of graft function and its histological features are transplant glomerulopathy, tubular basement membrane double contouring, C4d deposition or, alternatively, the presence of a molecular fingerprint of endothelial cells activation [3,4,5]. It is well-recognized that the main pathogenic mediators and the key diagnostic markers in this setting are donor specific antibodies (DSA), mainly, but not exclusively, recognizing human leukocyte antigens (HLA) antigens [6]. The fine molecular mechanisms underlying chronic antibody-mediated rejection are still largely unclear, despite the number of studies carried out in this field. This lack of knowledge complicates the identification of molecular targets clinically and pathologically relevant. Consequently, timely diagnosis and effective therapy for this condition are still far from the clinical routine.

Several studies described inflammatory cell types involved in renal allograft rejection [7], although most of the investigations focused on immune cells infiltrating the graft and potentially responsible of graft damage. A growing body of evidence suggests the importance of molecular analysis of peripheral blood mononuclear cells (PBMCs) in the attempt to understand the biological mechanisms that underlie the pathogenesis of antibody-mediated rejection. Our group characterized the molecular signature of PBMCs and CD4+ T lymphocytes isolated from patients with chronic antibody-mediated rejection, demonstrating the specific activation of interferon alpha signaling [8]. Sharbafi et al. reported that increased gene expression of toll-like receptor 4 (TLR4) in PBMCs could be suggestive of cell-mediated rejections [9]. Danger et al. described that miR-142-5p in circulating immune cells might represent a useful as a biomarker in antibody-mediated rejection [10].

In the last two decades, proteomic analysis has been largely applied in disease-biomarkers discovery and drug targets identification [11]. This approach has been largely applied also in the transplantation field in the attempt to recognize new sensitive and specific biomarkers of acute and chronic graft rejection [12,13,14]. The standard proteomic approach characterizes the proteins profile in a biological sample, but it gives only limited information on their functional activities. Protein phosphorylation modulates proteins’ functions and properties in most cellular processes [15]. Thus, its derangement is expected to contribute to the genesis and progression of diseases. On the other hand, most of phosphorylated proteins may be potential targets for disease drug therapy. Thus, the aim of our study was to analyze the PBMCs phosphoproteome, in the attempt to improve knowledge of the molecular mechanisms involved in chronic antibody-mediated rejection.

## 2. Results

Phosphoproteomic analysis of PBMC revealed different protein spots in antibody-mediated rejection vs. kidney transplant and healthy controls.

The main demographic, clinical, and histological features of the patients’ population are summarized in Table 1.

Five patients from antibody-mediated rejection, kidney transplant and healthy control groups were randomly selected for phosphoproteomic analysis. Phosphoproteins were isolated from total PBMCs’ proteins by precipitation with lanthanum chloride with an overall yield of 5.6 ± 4.4% for healthy subjects and of 3.6 ± 2.8% for transplant recipients. Phosphoproteins were, then, separated by two-dimensional gel electrophoresis (2DE), and a reference gel (phosphoproteome map) was generated for each of the three groups (CAMR, CTx, HC) by image analysis of Sypro Ruby-stained analytical gels. Spot analysis detected 462 ± 133 (mean ± SD) protein spots (CV = 26%) in healthy individuals, 465 ± 111 protein spots (CV = 23%) in patients with antibody-mediated rejection and 460 ± 90 protein spot (CV = 20%) in kidney transplant recipients with stable renal function. The 2DE reference gels from a patient with antibody-mediated rejection, a control kidney graft recipient and an immunocompetent control are shown in Figure 1A.

Differential protein spots analysis identified five protein spots with a significantly increased density in patients with chronic antibody-mediated rejection, compared to control transplant recipients and immunocompetent control subjects (fold change >2, Table 2). These spots corresponded to three proteins identified by mass spectrometry as lactotransferrin (Figure 1B,C, white box **a** in panel A), actin-related protein 2 (ARPC2) (Figure 1B,C, white box **b** in panel A) and calgranulin-B (Figure 1B,C, white box **c** in panel A) (Table 2).

### 2.1. Phosphorylation of ARPC2 Was Increased in Patients with Chronic Antibody-Mediated Rejection

To confirm the phosphoproteomic data we evaluated, by WB analysis, the expression of ARPC2 in PBMCs’ phosphoproteins isolated from patients and controls of the training set. We confirmed an increased expression of ARPC2 among phosphoproteins isolated by chronic antibody-mediated rejection patients compared to immunocompetent and kidney graft recipients controls (Figure 2A). In addition, we confirmed the increased expression of the phosphorylated ARPC2 in the test set of patients by double blotting of the total PBMCs’ proteins with the anti-ARPC2 antibody, followed by the anti phospho-serine antibody (Figure 2B).

### 2.2. ARPC2 Phosphorylation Is Correlated to Derangement of F-Actin Filaments in Patients with Chronic Antibody-Mediated Rejection

ARPC2 is one of the seven proteins that constitute the ARP2/3 complex, a key cytoskeletal regulator of actin polymerization, leadings to the formation of branched F-actin networks. In order to evaluate if ARPC-2 was directly involved into actin filaments organization in lymphomonocytes we evaluated, by confocal microscopy, the specific expression of ARPC2 and F-actin filament in patients with chronic antibody-mediated rejection. Interestingly, circulating T Lymphocytes isolated from graft recipients with chronic antibody-mediated rejection highly expressed ARPC2 in correspondence of F-actin filaments, which are recognized by phalloidin, (Figure 3A), while this association was weakly present in kidney transplant recipients with normal graft function (Figure 3B).

## 3. Discussion

To our knowledge, the present study represents the first analysis of PBMCs’ phosphoproteome in patients with chronic antibody-mediated kidney rejection. Protein phosphorylation is a key post-translational modification in the transmission deeply influencing protein functional activity. Most of the intracellular signaling pathways activated by cytokine, growth factors, cell to cell or cell to extracellular matrix interactions are based on protein phosphorylation. We standardized PBMCs’ phosphoproteome map of renal transplant recipients, which could be a useful source to investigate the changes in immune cells activation associated with different types of kidney transplant injury. It is known that two-dimensional gel electrophoresis analysis is quite labor intensive and time consuming, thus, differential protein analysis required, usually, a limited number of samples, which is then confirmed in a larger cohort by faster immunological protein analysis methods.

Phosphoproteome analysis identified three multifunctional proteins featuring chronic antibody-mediated rejection. It has already been recognized a role in the regulation of the immune response for lactoferrin [16,17] and calgranulin-B [18], while the increased expression and phosphorylation of ARPC2 is a novel finding in the potential pathogenic mechanisms of antibody-mediated chronic rejection. Interestingly, Nakorchevsky et al. conducted a large-scale proteogenomic study of kidney transplant biopsies with IFTA of varying severity, identifying 1400 proteins, with unique expression profiles, able to trace the progression from normal transplant biopsies to biopsies with mild to moderate and severe disease. Among them, a comprehensive control network for the actin cytoskeleton including several subunits of the actin-related protein 2/3 complex was found uniquely expressed (upregulated) in patient samples with progressive injury and a high risk for graft loss [19]. A recent proteomic analysis of allograft biopsies from renal transplant patients supported these findings by demonstrating that increased expression of ARP2/3 subunit 2 correlated with kidney fibrosis [20]. We further confirm their observation adding a brand-new piece of information; however, we do not know how cytoskeleton modification in PBMC could be correlated with increased fibrosis in the graft. Indeed, these studies [19,20] employed a classical proteomic approach identifying the proteins up or downregulated at the graft tissue level, whereas we investigated circulating immune cells using what we may define a functional proteomic approach. More functional studies including the recruitment of these cells in the kidney and their role in inflammation and fibrosis in the graft should be performed to better clarify the role of PBMC’s cytoskeleton modification in the pathogenesis of graft rejection.

Intriguing, our previous work on gene expression profile of PBMCs from CAMR patients, identified a molecular signature of 16 microRNAs (miRNA) downregulated in CAMR subjects compared to controls [8]. Among the 16 miRNA discriminating CAMR and control subjects, the expression level of miR-29b-3p was decreased in CAMR, and this miRNA is predicted to target ARPC2 (DIANA tools-MicroT-CDS. ENSG00000163466 (ARPC2), miTG score 0.889) [8]. Conventional Western blot analysis confirmed the increased expression of ARPC2 both in the PBMCs’ phosphoproteins of the training set of patients used for the proteomic analysis and in an independent set of patients, although we observed a variability in ARPC2 protein expression among patients due to personal physiological differences. ARPC2 is a serine (Ser108) phosphorylated protein [21,22] and could present multiple phosphorylation sites as reported in PhosphoSitePlus (http://www.Phosphosite.org). Double blotting of the total PBMCs’ proteins confirmed the increased expression of the phosphorylated ARPC2 in an independent set of CAMR patients compared to immunocompetent subjects and transplant recipients with stable graft function. ARPC2 is one of the seven proteins which constitute the ARP2/3 complex [23], a key cytoskeletal regulator of actin polymerization, a critical process for the protrusion of cell membrane, which is fundamental for cell motility. The activity of the ARP2/3 complex comprises a variety of cellular functions, including change of cell shape, motility, endocytosis, and phagocytosis [24]. The ARP2/3 complex initiates new actin filaments from the side of preexisting ones leadings to the formation of branched F-actin networks, which commonly form the structure of the lamellipodium, essential for cellular motility [25]. Confocal microscopy analysis not only confirmed the overexpression of ARPC2 in PBMCs of patients with antibody-mediated rejection compared to healthy controls, but demonstrated also the association between ARPC2 and F-actin. An association between the actin family cytoskeletal proteins and F-actin-based structures was found in quantitative proteomic analysis and dynamic mapping of calcineurin inhibitors (CNI)-exposed porcine proximal tubular proteome [26]. The study demonstrated the effect of cyclosporine on actin remodeling in particular, the cyclosporine exposure of renal proximal tubular cells sustained a significant reorganization of cortical actin cytoskeleton related to a significant loss of F-actin-based structures. We wondered whether cyclosporine treatment might be a bias in interpreting our data although the cell type described in our paper is completely different, and only less than half of our cohort of patients received that treatment. The generation of new actin filaments by nucleation is a critical checkpoint in the control of actin polymerization, however, the Arp2/3 complex is an inefficient nucleator on its own. Its activity is usually stimulated by nucleation-promoting factors such as Wiskott-Aldrich syndrome protein (WASP) family proteins [27]. WASP, uniquely expressed in hematopoietic cells, including most of the circulating immune cells (dendritic cells, macrophages, T, B and natural killer lymphocytes), plays a key role in the regulation of cytoskeleton organization [28] and in cell signaling [29]. WASP induces actin polymerization which is a key feature of T and B cell receptor activation [30,31]. Indeed, this nucleating protein is crucial for effector T and, particularly, B cell activation. Lack of WASP induces a condition of humoral immune deficiency mainly characterized by dysgammaglobulinemia and a reduced response to vaccination [32,33]. WASP in immune cells serves as an integrator between cell surface signaling cascades and the actin cytoskeleton network, which is vital for an efficient immune response. In fact, the actin apparatus enables T cells to migrate and invade inflamed tissues, where they encounter and interact with antigen-presenting cells (dendritic cells, macrophages, and B cells) priming their antigen-specific activation [34]. In addition, WASP is essential for B-cell development in the bone marrow, it has a key role in B-cell homeostasis [35], and its deficiency influences B-cell adhesion, migration and homing, delaying the humoral immune response [33]. In our setting, it is conceivable that an activated WASP signaling cascade with the subsequent changes in cytoskeleton organization may play a pathogenic role in B cell activation featuring antibody-mediated rejection.

The main limitations of our study are the small number of patients enrolled. Obtained results need to be confirmed in a larger cohort of patients and other functional evidences of the specific effect of altered cytoskeleton organization need to be investigated. It has been described that T cell polarity and motility, play an important role in the interaction with antigen presenting cells [36] and DOCK2 deficiency in mice, a regulator of the actin cytoskeleton in lymphocytes, suppresses cardiac allograft rejection in the recipients by reducing priming and activation of naive T cells and by attenuating graft infiltration of activated T cells [37]. The influence of altered ARPC2 expression and phosphorylation on the progression of renal damage in CAMR and the mechanisms that can correlate this process identified in PBMC with progression of fibrosis in the graft may represent further interesting items to investigate.

In conclusion, our results are the first evidence of altered cytoskeleton organization in circulating immune cells of patients with chronic antibody-mediated rejection. The increased expression of phosphorylated ARPC2 found in the PBMCs of our patients and its association with derangement of F-actin filaments might suggest that proteins regulating actin dynamics in immune cells could be involved in the mechanism of chronic antibody-mediate rejection of kidney grafts.

## 4. Materials and Methods

### 4.1. Materials

Acetonitrile (ACN), acetone, trifluoroacetic acid (TFA), trichloroacetic acid (TCA), DL-dithiothreitol (DTT), iodoacetamide (IAA), glycine, EDTA, Tris, endonuclease, phosphoprotease and protease inhibitors, lanthanum chloride, potassium dihydrogen phosphate, Coomassie Blue G-250 were purchased from Sigma (Sigma-Aldrich, St. Louis, MO, USA); urea, CHAPS, SDS, glycerol, acrylamide, ampholine, and Ficoll-Paque™ were purchased from GE Healthcare (Uppsala, Sweden); Agarose, Pro-Q^®^ Diamond dye, SYPRO^®^ Ruby and PeppermintStick™ Phosphoprotein Molecular Weight Standards were from Invitrogen™ (Carlsbad, CA, USA); piperazine di-acrylamide (PDA), TEMED, Bio Rad Protein Assay, IPG strips, were from Bio-Rad Laboratories (Hercules, CA, USA). Trypsin (sequencing grade modified) was from Promega (Madison, WI, USA). All solvents used were Ultra-Resi-Analyzed grade.

### 4.2. Patients

We analyzed PMBCs samples from 38 individual: 16 kidney transplant recipients with clinical and histological evidence of chronic antibody-mediated rejection, 11 kidney transplant recipients with normal graft function and histology, and 11 healthy immunocompetent individuals as control groups (Table 1). After signing informed consent, twenty-seven kidney transplant recipients, undergoing a graft biopsy, were included in the study in agreement with the following inclusion criteria: age > 18 years; biopsy-proven chronic active antibody-mediated rejection according to Banff 2011 criteria [38]; calcineurin inhibitor-based immunosuppressive therapy at the time of enrolment; absence of histological signs of acute rejection, systemic infection, systemic inflammatory diseases, and known or clinically suspected malignancies. Kidney transplant recipients undergoing protocol graft biopsies, with normal renal function and histology, in the absence of circulating anti-HLA antibodies, represented the control group (*n* = 11). Eleven immunocompetent subjects were ascertained as healthy by a general medical examination, they were not taking drugs, and signed informed consent. Five patients from each group were randomly selected for phosphoproteomic analysis (training set). PBMCs of 26 patients (test set) were used for proteomic’s results validation. The study was conducted according to the latest version of the Declaration of Helsinki and was approved by the local ethics committee (Prot. No. 670/C.E., 30.04.2014).

### 4.3. PBMCs Isolation and Total Proteins Extraction

PBMCs were isolated from whole blood (25 mL) as previously described [39]. Total proteins were extracted by adding 500 μL RIPA buffer with 10 μL endonucleases, and 5 μL phosphoprotease and protease inhibitors to PBMCs pellet (4 × 10^7^) as previously described [39]. Approximately 1.3 (1.29 ± 0.3) mg and 1.5 (1.56 ± 0.62) mg of total proteins were obtained from PBMCs of kidney transplant recipients and immunocompetent controls, respectively.

### 4.4. Phosphoproteins Isolation

Phosphoproteins were isolated from total PBMCs’ proteins by precipitation with lanthanum chloride, as previously described [39].

Two-Dimensional Gel Electrophoresis (2DE) and Matrix-Assisted Laser Desorption Ionization-Time of Flight Tandem Mass Spectrometry (MALDI-TOF MS/MS) Analysis.

Phosphoproteins isolated by lanthanum chloride were separated by 2D-PAGE and identified by MALDI-TOF MS/MS analysis as previous described [39]. ImageMaster™ 2D Platinum software (Amersham Biosciences, Little Chalfont, UK) was used for image analysis of analytical 2DE gels as already described [40]. The fold change of protein expression among different classes (chronic antibody-mediated rejection versus control kidney graft recipients and immunocompetent subjects) was calculated considering the mean of spot intensity (measured as the relative volumes of spots) of the reference gels in each class [40]. All 2DE gels were run in triplicate.

### 4.5. Western Blot Analysis

Phosphoproteomic data were confirmed by Western blot (WB) analysis both, in the training set of patients used for proteomic analysis and in a testing set including 23 different samples (11 chronic antibody-mediated rejection, 6 control transplant recipients, and 6 immunocompetent patients). PBMCs’ phosphoproteins (10 μg) isolated from the training set were separated by one dimensional (1D) SDS gel electrophoresis. After transfer, the membrane was blocked in 5% BSA (PBS-T 0.1%), and then incubated overnight with primary monoclonal anti-ARPC2 antibody (Abcam, Cambridge, UK). Membranes were incubated with the appropriate HRP-conjugated secondary antibody and acquired after the addition of the proper substrate. In the same way, total PBMCs’ proteins (100 μg), isolated from the testing set, were analyzed by WB as described above. After the incubation with primary monoclonal anti-ARPC2 antibody and image acquisition, the membranes were stripped and immunoblotted again with monoclonal anti phospho-Ser antibody (Abcam). Clarity™ Western ECL substrate (Bio-Rad, Hercules, CA, USA) was used to detect protein bands by the Versadoc Molecular ImagerTM (Bio-Rad). Results of densitometry analysis were expressed in arbitrary units. Data were normalized to the total protein content in each sample. The immunoblotting experiments were run at least in duplicate.

### 4.6. Confocal Microscopy

The expression and interaction of ARPC2 and F-actin filaments were evaluated in vitro on PBMCs of 3 patients with antibody-mediated rejection and 3 control kidney transplant recipients by double-fluorescence immune-labelling and confocal microscopy. PBMCs (106/slide) were seeded on coated poly-D-lysine (Sigma-Aldrich, St. Louis, MO, USA) cover glass. The cells were then fixed with paraformaldehyde 4% in PBS (Sigma-Aldrich, St. Louis, MO, USA), treated with Triton 0.1% for 5 min and incubated for 1 h with a blocking solution of bovine serum albumin 2% in PBS. Subsequently, the cells were incubated with a monoclonal anti-ARPC2 antibody (Abcam. 1:1000 dilution) for 1h and, after washing, with the specific secondary antibody IgG2-Alexa Fluor 488 conjugate (1:200 dilution Molecular Probes, Eugene, OR, USA) for 1 h at room temperature. After further 30 min incubation with the blocking solution, cells were incubated with Phalloidin Oregon green 514 (Invitrogen-Thermo Fisher, Carlsbad, CA, USA) for 20 min. The slides were then mounted in Gel/Mount (Biomeda, Foster City, CA, USA) and sealed.

The slides were examined under a fluorescence microscope equipped with appropriate filters (Leica TCS SP2, Leica Microsystems, Wetzlar, Germany). Images from individual optical planes and multiple serial optical sections were analyzed, and the images were sequentially scanned. Image analysis was performed on all acquired fields. Negative secondary antibody controls were processed in parallel.

### 4.7. Statistical Analysis

The results of the quantitative variables were expressed as mean ± SD, unless otherwise indicated. Differences between quantitative variables were tested by the Mann Whitney U-test and Kruskall Wallis ANOVA, as appropriate. *p* values < 0.05 were considered statistically significant. The Statview software package, SAS (v. 5.0) was used for all analyses.

## Figures and Tables

**Figure 1 ijms-21-06509-f001:**
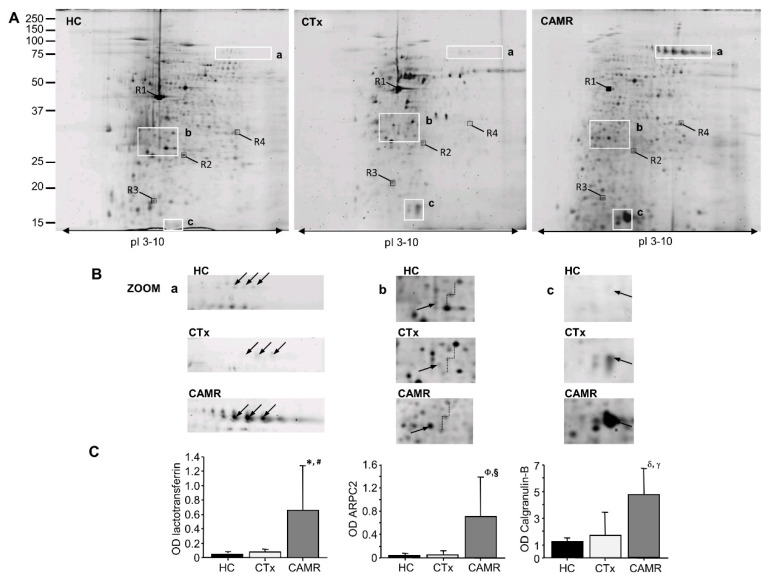
(**A**) Two-dimensional gel electrophoresis (2DE) peripheral blood mononuclear cells (PBMCs) phosphoprotein reference maps of healthy subjects (HC), transplanted patients with stable renal function (CTx) and chronic antibody-mediated rejection (CAMR) patients. Molecular weight and pI values are indicated on the left side and at the bottom of the image. The areas with the most relevant differences in the protein expression among groups are evidenced in white boxes (a, b, c). (**B**) matrix-assisted laser desorption ionization-time of flight tandem mass spectrometry (MALDI-TOF-MS/MS) identified spots in box a as lactotransferrin (LTF) (*: *p* = 0.01 vs. CTx; #: *p* = 0.01 vs. HC); spots in box b as actin-related protein 2 (ARPC2) (Φ: *p* = 0.0005 vs. CTx; §: *p* = 0.0006 vs. HC); spots in box **c** as calgranulin-B (S10A9) (δ: *p* = 0.03 vs. CTx; γ: *p* = 0.02 vs. HC). (Fisher’s protected least significant difference for ANOVA test). (**C**) The semiquantitative variation in protein expression measured by densitometric analysis of spots from the same sample groups considering the relative volumes (vol%) of spots.

**Figure 2 ijms-21-06509-f002:**
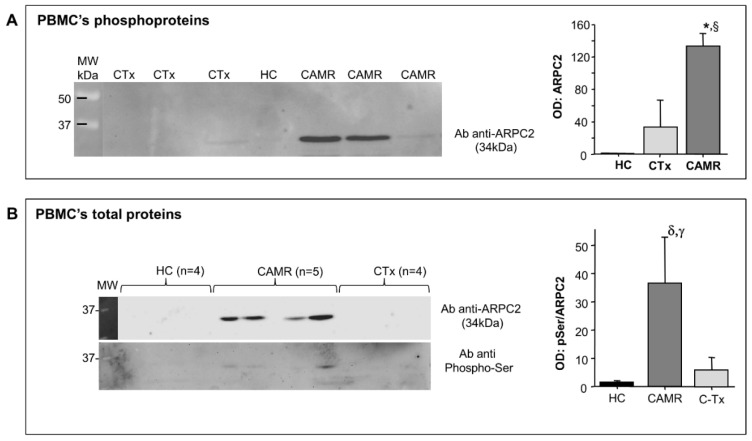
(**A**) A representative Western blot analysis of PBMCs’ phosphoproteins isolated from normal and pathological samples by lanthanum chloride. The statistically significant increase of ARPC2 in CAMR patients compared to CTx is highlight in the graphic at the right. Results are expressed as arbitrary units (OD: optical density) and normalized to the total amount of phosphoprotein loaded (*: *p* = 0.03 vs. CTx; §: *p* = 0.04 vs. HC). (Fisher’s PLSD for ANOVA test); (**B**) A representative Western blot of total PBMCs’ proteins extracted from HC, CAMR and CTx patients. Results are expressed as arbitrary units (OD: ratio between of optical density for anti-phosphoserine and optical density for anti-ARPC2) and normalized to the total amount of protein loaded. (δ: *p* = 0.04 vs. CTx; γ: *p* = 0.02 vs. HC). (Fisher’s PLSD for ANOVA test).

**Figure 3 ijms-21-06509-f003:**
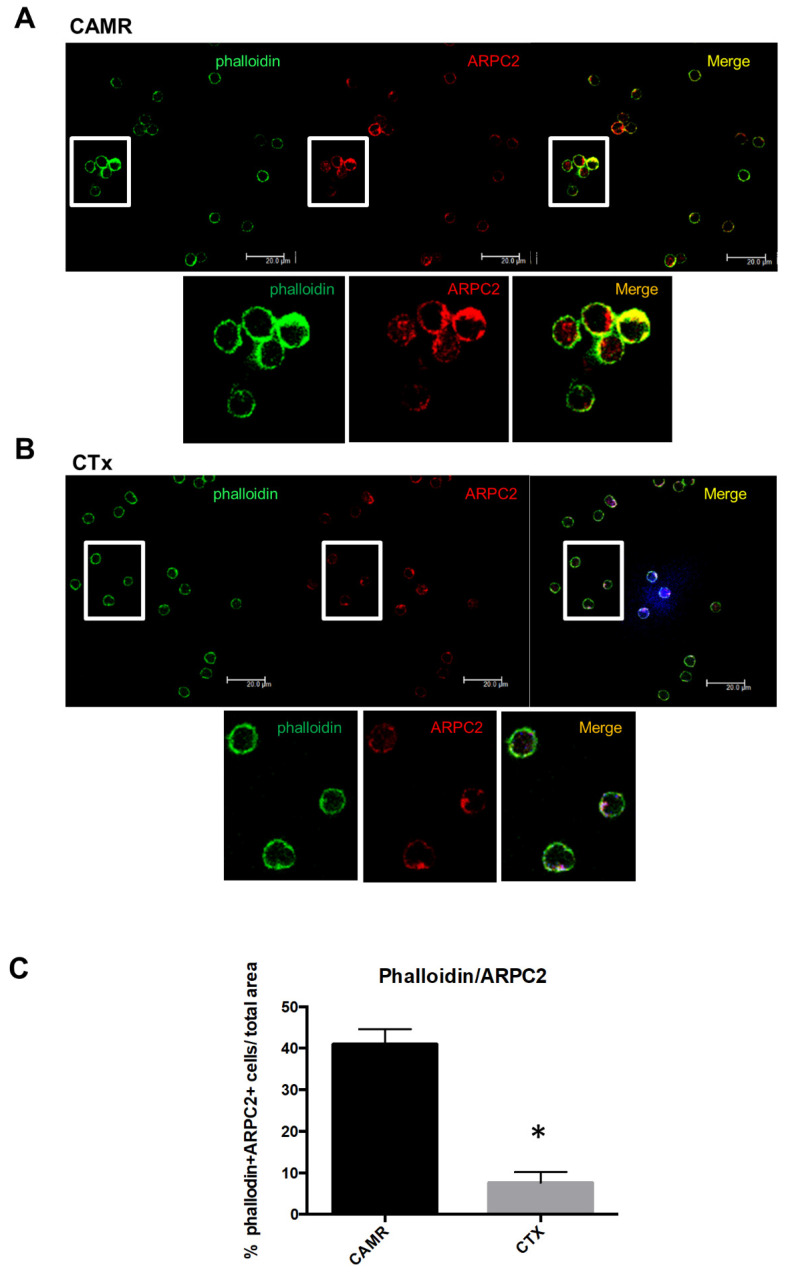
Confocal laser microscopy analysis of ARPC2 and F-Actin protein expression in PBMCs of CAMR patients (**A**) and a kidney transplant recipients with normal graft function (CTx) (**B**). Quantification of the percentage of falloidin and ARPC2 positive cells (**C**). * *p* < 0.05.

**Table 1 ijms-21-06509-t001:** Demographics, clinical and histological characteristics of the enrolled patients.

	CAMR-GROUP	CTRL-GROUP	*p*-Value
Number (patients)	16	11	
Age (years)	44.4 ± 8.7	53.1 ± 14.2	*NS*
Gender (M/F)	9/7	6/5	
Time since transplantation (months)	7 ± 5.8	8.9 ± 6.9	*NS*
Serum creatinine (mg/dl)	1.9 ± 0.6	1.2 ± 0.4	*0.02*
Proteinuria (g/24h)	2.6 ± 2.5	0.42 ± 0.7	*0.01*
Cyclosporine	7	3	-
Tacrolimus	9	8	-
Number of mismatches HLA	3 ± 0.6	2.5 ± 0.5	*NS*
Anti HLA antibodies (%)	100%	15%	-
Banff Score Chronic Glomerulopathy	0: 38%; 1: 9%; 2:15%; 3: 38%	0: 100%	-
Banf Score Peritubular Capillaries	0: 65%; 1: 14%; 2: 21%; 3: 0%	0: 100%	-
Glomerulitis Banff Lesion Score	0: 45%; 1: 25%; 2: 20%; 3: 10%	0: 100%	-
Interstitial Fibrosis Banff Lesion Score	0: 0%; 1: 40%; 2: 50%; 3: 10%	0:80%; 1: (20%)	-
Tubular atrophy Banff Lesion Score	0: 0%; 1: 41%; 2: 50%; 3: 9%	0:80%; 1: (20%)	-
C4d positive	65%	0%	

**Table 2 ijms-21-06509-t002:** Phosphorylated PBMCs’ proteins differentially expressed in CAMR patients, compared to transplanted patients with stable renal function (CTx) and healthy controls (HC).

Protein Name (GENE Name)	Accession Number	Molecular Weight (Da)	Mascot Score	Seq. Cov. (%)	*P* Value ^a^ CAMR vs. CTx (Fold Change)	*P* Value ^a^ CAMR vs. HC (Fold Change)
Lactotransferrin (LTF)	P02788	80014	257	41	0.01 (10.1)	0.01 (19.1)
Actin-related protein 2/3 complex subunit 2 (ARPC2)	O15144	34333	97	30	0.0005 (3.4)	0.0006 (3.2)
Calgranulin-B (S10A9)	P06702	13291	122	79	0.03 (2.7)	0.02 (3.8)

^a^: Fisher’s PLSD for ANOVA test (*p* < 0.05).

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
