# Peer review of "Altered Phosphorylation of Cytoskeleton Proteins in Peripheral Blood Mononuclear Cells Characterizes Chronic Antibody-Mediated Rejection in Kidney Transplantation"

_ijms, 2020, doi:10.3390/ijms21186509_

Round 1
Reviewer 1 Report
By performing phosphoproteome analysis of peripheral blood mononuclear cells authors found an altered cytoskeleton organization in circulating immune cells of kidney transplant recpients with chronic antibody-mediated rejection. Significant immunopathological consequences associated with genetic alteration of actin cytoskeletal regulatory genes are well described in other pathologies. Altough interesting the results still are preliminary and require subsequent verification. The work is limited by being from a single center without an external validation cohort.
Author Response
The authors wish to thank the Reviewers and the Editorial Committee for the careful and accurate revision of the original manuscript, as well as for their suggestions and criticisms.
Reviewer 1
Comments and Suggestions for Authors
By performing phosphoproteome analysis of peripheral blood mononuclear cells authors found an altered cytoskeleton organization in circulating immune cells of kidney transplant recpients with chronic antibody-mediated rejection. Significant immunopathological consequences associated with genetic alteration of actin cytoskeletal regulatory genes are well described in other pathologies. Altough interesting the results still are preliminary and require subsequent verification. The work is limited by being from a single center without an external validation cohort.
Response:We agree with this comment by the reviewer. In the discussion of the manuscript we already reported these limitations of our study. We are also aware that other functional evidences of the specific effect of altered cytoskeleton organization need to be investigated.

Reviewer 2 Report
This is a nice paper. However, I have some comments.
The findings from this paper are excellent and worthy to review.
This manuscript contained some questions described below.
I think this paper is interesting, this review contributes to future's clinical medicine largely. I have some questions from a point of view of clinical medicine.
In the CAMR group, there seem to be many patients receiving cyclosporine. Is this a bias in interpreting the data?
The CAMR group seems to have many interstitial fibrosis and C4d positivity, and less glomerulitis and peritubular capillaries. Is this related to ARPC2 or lactotransferrin positivity? Since tubular injury is thought to be significantly associated with renal prognosis, please add data and discussion regarding the relationship between this study and histopathological findings.
What is the relationship with clinical markers for tubular injury? Please tell me the relationship between urine L-FABP and β2 microglobulin. Or add some thought.
Author Response
Point-by-point answers to Reviewers
The authors wish to thank the Reviewers and the Editorial Committee for the careful and accurate revision of the original manuscript, as well as for their suggestions and criticisms.
Reviewer 2
Comments and Suggestions for Authors
This is a nice paper. However, I have some comments.
The findings from this paper are excellent and worthy to review.
This manuscript contained some questions described below.
I think this paper is interesting, this review contributes to future's clinical medicine largely. I have some questions from a point of view of clinical medicine. In the CAMR group, there seem to be many patients receiving cyclosporine. Is this a bias in interpreting the data?
Response:This is an interesting question. In our study 10 over 27 patients belonging to both groups, CAMR as well as kidney transplant recipients with normal graft function, received cyclosporine, therefore, a possible effect of cyclosporine on the data obtained would still be partial. An effect of cyclosporine on actin remodelling was described in proximal tubular cells by Burat and co-workers (Burat B, et al. Cyclosporine A inhibits MRTF‐SRF signaling through Na+/K+ ATPase inhibition and actin remodeling. FASEB BioAdvances. 2019; 1:561–578). They performed the quantitative proteomic analysis and dynamic mapping of Calcineurin Inhibitors (CNI)‐exposed porcine proximal tubular proteome and demonstrated that Cyclosporine A displayed significant over‐representation and differential expression of actin family cytoskeletal proteins in particular a significant loss of F‐actin‐based structures. We introduced this point in the discussion, although the cell type described in our paper is completely different.
The CAMR group seems to have many interstitial fibrosis and C4d positivity, and less glomerulitis and peritubular capillaries. Is this related to ARPC2 or lactotransferrin positivity?
Response:Since tubular injury is thought to be significantly associated with renal prognosis, please add data and discussion regarding the relationship between this study and histopathological findings.
It has been recently demonstrated in kidney allograft biopsies that increased expression of Actin-related protein 2/3 (ARP2/3) subunit 2 correlated with kidney fibrosis (Mortensen LA, Svane AM, Burton M, et al. Proteomic Analysis of Renal Biomarkers of Kidney Allograft Fibrosis-A Study in Renal Transplant Patients. Int J Mol Sci. 2020;21(7):2371.), thus supporting data already described by Nakorchevsky et al. (ref.19 in the paper) who identified the actin-related protein 2/3 complex uniquely expressed (upregulated) in patient samples with progressive injury and a high risk for graft loss. Despite these results, we don’t know how cytoskeleton modification in PBMC could be correlated with increased fibrosis in the graft. More functional studies including the recruitment of these cells in the kidney and their role in inflammation and fibrosis in the graft should be performed, but are actually behind the scope of our paper. In the revised version of the manuscript we also included the paper by Mortensen et al and inserted a comment on this point in the discussion.
What is the relationship with clinical markers for tubular injury? Please tell me the relationship between urine L-FABP and β2 microglobulin. Or add some thought.
Response:We thank the reviewer for these suggestions. Actually the scope of our paper was to identify cellular signalling networks differentially activated in CAMR patients through a phosphoproteomic approach, that, in our knowledge, has never been applied in the study of CAMR. Further studies on a larger cohort of patients from different centers will allow us to understand if altered ARPC2 expression and phosphorylation might really influence the progression of renal damage in CAMR and which are the mechanisms that can correlate this process identified in PBMC with progression of fibrosis in the graft. In the novel version of the manuscript we introduced this point as a limitation of the study.

Round 2
Reviewer 2 Report
I read the paper with great interest.
This paper is well written and informative
I have no specific comments.